# Celebrating 40 Years of Ironman: How the Champions Perform

**DOI:** 10.3390/ijerph16061019

**Published:** 2019-03-20

**Authors:** Lucas Pinheiro Barbosa, Caio Victor Sousa, Marcelo Magalhães Sales, Rafael dos Reis Olher, Samuel Silva Aguiar, Patrick Anderson Santos, Eduard Tiozzo, Herbert Gustavo Simões, Pantelis Theodoros Nikolaidis, Beat Knechtle

**Affiliations:** 1Graduate Program in Physical Education, Catholic University of Brasília, 71966-700 Brasília, Brazil; lduarte.barbosa@gmail.com (L.P.B.); cvsousa89@gmail.com (C.V.S.); rflolher@gmail.com (R.d.R.O.); ssaguiar0@gmail.com (S.S.A.); patricksantospas@gmail.com (P.A.S.); hgsimoes@gmail.com (H.G.S.); 2Miller School of Medicine, University of Miami, Coral Gables, FL 33124, USA; etiozzo@med.miami.edu; 3Physical Education Department, Goias State University, Quirinópolis, 75860-000 GO, Brazil; marcelomagalhaessales@gmail.com; 4Exercise Physiology Laboratory, 18450 Nikaia, Greece; pademil@hotmail.com; 5Medbase St. Gallen Am Vadianplatz, 9001 St. Gallen, Switzerland; 6Institute of Primary Care, University of Zurich, 8006 Zurich, Switzerland

**Keywords:** triathlon, cycling, running, swimming, endurance

## Abstract

We aimed to determine which discipline had the greater performance improvements in the history of Ironman triathlon in Hawaii and also which discipline had the greater influence in overall race time. Data from 1983 to 2018 of the top three women and men of each year who competed in the Ironman World Championship were included. In addition to exploratory data analyses, linear regressions between split times and years of achievement were performed. Further, a stepwise multiple linear regression was applied using total race time as the dependent variable and split times as the independent variables. Both women and men significantly improved their performances from 1983 to 2018 in the Ironman World Championship. Swimming had the largest difference in improvements between men and women (3.0% versus 12.1%, respectively). A negative and significant decrease in each discipline was identified for both women and men, with cycling being the discipline with the greatest reduction. The results from the stepwise multiple regression indicated that cycling was the discipline with the highest influence on overall race time for both sexes. Based on the findings of this study, cycling seems to be the Ironman triathlon discipline that most improved overall race times and is also the discipline with the greatest influence on the overall race time of elite men and women in the Ironman World Championship.

## 1. Introduction

The Ironman triathlon consists of swimming 2.4 miles (3.8 km), cycling 112 miles (180 km), and running 26.2 miles (42.2 km) and is considered as one of the most challenging ultra-endurance events worldwide [1,2]. Although triathlon started in San Diego, California, the history of Ironman triathlon started in 1978 in Hawaii, with the first Ironman World Championship being held in Kailua-Kona (Big Island) three years later, also in Hawaii [1,2,3].

Nowadays, the Ironman events take place all over the world, with amateur and professional athletes competing in these events to qualify for the World Championship in Kailua-Kona. Ironman Hawaii in considered as the toughest Ironman race in the world due to the course, the environmental conditions, and the competitiveness of the event [2,4]. The race itself is one of the most popular triathlon events in the world, with a growing competitiveness and performance improvement in non-elite triathletes [1,5,6]. In addition, it should be highlighted that the best professional triathletes in the world often achieve new records in Kailua-Kona [7].

In order to help coaches and athletes with both training plans and race strategy, performance trends have been analyzed in the past few years in many endurance sports [8,9,10,11]. Specifically in triathlon, relevant studies have been conducted for Olympic distance (1.5 km swim/40 km cycle/10 km run) [12,13], half-distance (half-Ironman: 1.9 km swim/90 km cycle/21 km run) [13,14], full-distance (3.8 km swim/180 km cycle/42.195 km run) [14,15], and ultra-triathlons (distance > Ironman) [16,17]. To date, two studies investigated the performance of amateur triathletes [2,5], but none of them included only elite women and men.

Ofoghi et al. [18] investigated which discipline would have the greater influence on overall performance in an Olympic triathlon and concluded that running was the most decisive, followed by swimming and cycling. On the other hand, Sousa et al. [19] analyzed all sub-8-h performances in full-distance triathlon (i.e., Ironman) and reported that cycling was the discipline with the greatest influence on the overall result, followed by running and swimming. Additionally, it is noteworthy that in 2018 the female and male winners of the 2018 World Championship improved the course records, showing that the fastest Ironman triathletes worldwide can further improve their performances.

However, to the best of our knowledge, the only two studies analyzing the Ironman World Championship results concerned amateur athletes in the analysis, with one of the studies analyzing races up to 2007 [2] and the other analyzing races from 2002 to 2015 [5]. Therefore, we aimed to analyze only elite men and women competing in the Ironman World Championship from 1983 to 2018 in order to determine (i) which discipline had the greatest performance improvement in the last 35 years; (ii) which discipline had the greatest influence on overall result; and (iii) whether women were really closing the gap to men.

## 2. Methods

### 2.1. Ethical Approval

All procedures used in the study were approved by the Institutional Review Board of Kanton St. Gallen, Switzerland, with a waiver of the requirement for informed consent of the participants given the fact that the study involved the analysis of publicly available data (1 June 2010).

### 2.2. Data

All data were obtained from a publicly available database (www.ironman.com). All official overall race and split times from the top three women’s and men’s finishers of the Ironman World Championship from 1983 to 2018 were included in the analysis. Table 1 presents the descriptive distribution of women’s races including the Ironman World Championship Race/Split Record (among top three finishers), whereas Table 2 presents men’s data.

### 2.3. Statistical Analysis

Initially, an exploratory analysis of the data was carried out, in which central tendency (median and mean), dispersion (interquartile ranges (25 and 75 percentiles and standard deviation), and extreme (lowest value) measures were calculated (Table 1 and Table 2). Furthermore, all data were transformed in seconds and non-linear regressions (second order) were performed between each split time and year of achievement. Linear regressions were used for splits because the non-linear fitting line was the same as the linear. The relative difference (percentage) between the first (1983) and last (2018) World Championship’s top three performances was calculated for both women and men. Regarding overall race time, non-linear regression analyses were performed since the trend line had a better fit than linear regression. A comparison of average race times between the top three athletes and the chasing group (4th to 10th place finishers) was performed. Finally, a stepwise multiple linear regression was performed using overall race time as the dependent variable and split times as independent variables. The significance level was set as 5% (*p* < 0.05), and all procedures were performed using SPSS v21.0 (IBM SPSS Statistics for Windows. Armonk, NY: IBM Corp).

## 3. Results

Men improved in overall race time by 13.3% from 1983 to 2018, whereas women improved by 20.8% (Table 3). Swimming showed the largest difference in improvements between men and women (3.0% versus 12.1%, respectively), and running showed the smallest difference (12.5% versus 15.5%, respectively) for the three split disciplines.

Both women and men significantly improved their performances from 1983 to 2018 in the Ironman World Championship in Kona, Hawaii (Figure 1 and Figure 2). The world record was improved almost every three years (see Appendix A for accurate race time values from each year’s champions).

The linear regression of split disciplines shows a negative and significant slope for all disciplines for both women (swimming: −6.94 to 0.47; cycling: −71.06 to −36.98 *; running: −45.79 to −26.86 *; Figure 3) and men (swimming: −19.37 to −6.77 *; cycling: −96.01 to −60.52 *; running: −65.06 to −30.58 *; Figure 4) (* indicates *p* < 0.001). The greatest slope in both sexes was for cycling.

The best-fitting model from the stepwise multiple regression included swimming, cycling, and running split times for both women and men (Table 4). Cycling was the discipline with the greatest standardized beta for both sexes. The swimming discipline resulted in a negative standardized coefficient for the men.

## 4. Discussion

The main finding of this manuscript was that cycling has been the Ironman triathlon discipline with the greatest improvement rate throughout the years and also has had the greatest influence on overall race time for both women and men. However, apparently both women and men have improved their performances over the years in all triathlon disciplines. It is worth mentioning that women had a greater improvement than men in all triathlon disciplines and consequently in total race times. 

Jeukendrup and Martin [20] had previously reported that cycling in aero position and the use of lighter wheels (i.e., elbows on handlebars and carbon wheels, respectively, which had developed for use in time-trial and triathlon bicycles) makes an athlete significantly faster. Thus, cycling performance also had new technologies that could influence the performance increase, from the outfit to the bicycle itself, all of which contributed to make the athlete more comfortable, aerodynamic, and consequently faster.

Although cycling is the discipline that encompasses more time in comparison to swimming and running in Olympic distance and short distances, it does not have an influence in overall race time, being the least important of the three disciplines [21]. In Olympic distance and short distances, athletes normally swim really fast to be able to leave transition one with the first pack of cyclists and stay within the leading and chasing peloton, thus saving the energy for the running [18]. However, in Ironman races drafting during cycling is not allowed, making cycling a more competitive discipline, which means that athletes have to apply some strategy in order to cycle fast enough to remain in a competitive position but still save energy for the running leg. Similarly, in an analysis only including top full-distance triathlon performances, the authors reported that cycling was the discipline that most influenced overall performance in elite men racing below 8 h of overall race time, followed by running and swimming [19].

A performance analysis on Ironman races investigated more than 340,000 triathletes racing in 253 different race locations and concluded that the race tactics in an Ironman triathlon should focus on saving energy during the first two disciplines for the running split [22]. This conclusion is different from the findings of the present study, which suggest that athletes seem to apply greater effort in cycling than during running. It is worth mentioning that this analysis was carried out with a majority of age groupers (non-elite), whereas the present study only considered the top three elite professionals from each year. It is noteworthy that 4th to 10th place finishers in the Ironman World Championship seemed to have a substantial performance improvement in the last decade of the event, with consistently much closer groupings in the top ten athletes for both men and women.

With regard to performance throughout the race, the performance analysis of the Ironman World Championship with amateurs reported a performance increase in all disciplines for men and women [23]. However, the authors suggest that this improvement in performance may be due to an increased number of athletes and morphological changes [23]. The overall performance increase throughout the years can be mostly attributed to the development of new nutrition and training strategies [24,25,26,27]. A controversial result was the negative coefficient for the swim split in men, which would mean that a slower swim could lead to a better overall race time. We believe that this statistical outcome is due to the specificity of the sample, as only the top three athletes in the overall race were considered, and these athletes are not always the best swimmers. For example, in the 2018 World Championship, none of the top three overall athletes were among the top 10 swimmers.

Concerning the performance gap between men and women, it has markedly reduced in the last decades. At the 2018 Ironman World Championship, women improved by 21% while men improved by 13%; the absolute gap between them reduced from 1 h and 38 min to 33 min. Indeed, in the 2018 Championship, the female champion Daniela Ryf crossed the finish line ahead of 20 elite professional men who finished the race. Some previous studies have concluded that women have been closing the gap in swimming [28,29], in running [30,31], and even in triathlon [2]. We believe that women may still close the gap in an Ironman someday despite the body composition and physiological differences that exist between men and women. One of the possible explanations for this can be attributed to cultural changes that have favored a greater participation of women in all sports, including triathlon, thus increasing competitiveness and therefore performance [7,9,10,32].

When comparing performance with other endurance modalities such as ultra-triathlon and marathon running, the performance gaps between women and men are getting smaller each year. Knechtle et al. [33] investigated the performance trends of Double Iron ultra-triathlon (2I; 2x Ironman distance), Triple Iron ultra-triathlon (3I; 3x Ironman distance), and Deca Iron ultra-triathlon (10I; 10x Ironman distance) from 1985 to 2009 and reported a smaller sex difference in 2I and 3I. Conversely, Nikolaidis et al. [31] investigated the performance of male and female athletes running the marathon and concluded that men are still faster than women, but the performance gap remained unchanged for the past few years.

Regarding the specific Ironman World Championship, Kailua-Kona is one of the toughest races within the entire Ironman circuit, which typically requires athletes to swim in choppy waters, cycle with a lot of wind, and run in hot and sunny weather [34,35]. The race course has not always been the same in Ironman Hawaii, with small changes every two or three years in order to accommodate safety precautions and/or local transit logistics. Although this may affect the overall race time, the distances remained standard and we believe that any small course changes affecting a specific split race time are diluted within the sample and do not represent a great confounder to the general results of this study.

## 5. Conclusions

In conclusion, cycling seems to be the triathlon discipline that most improved overall race times and is also the discipline that had the greatest influence on the overall race time in elite men and women in the Ironman World Championship. Furthermore, within the last 40 years of Ironman Hawaii, both men and women improved overall time performance, but women improved more, thereby closing the gap to men.

## Figures and Tables

**Figure 1 ijerph-16-01019-f001:**
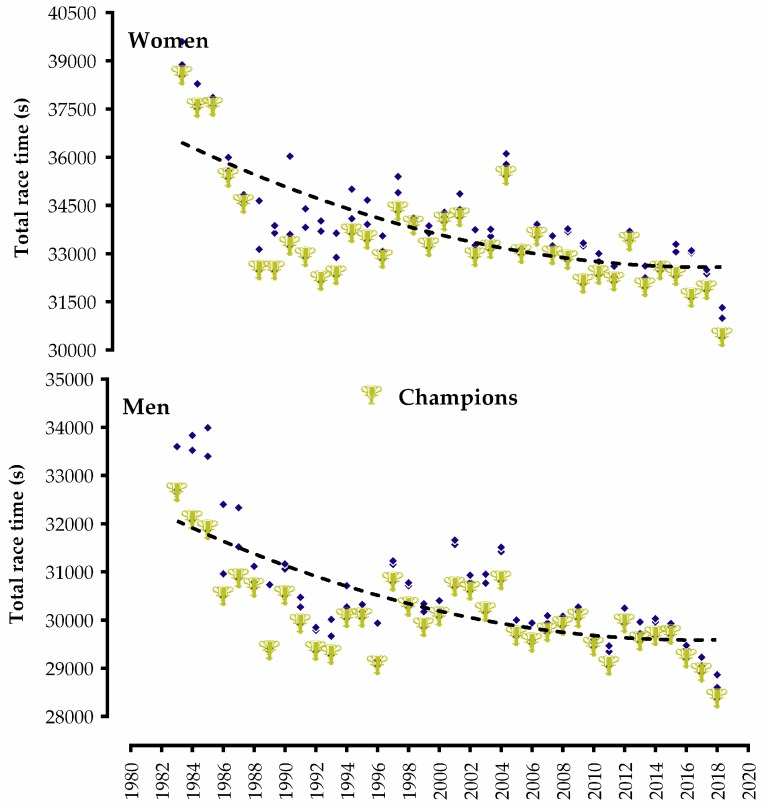
Dispersion and non-linear regression of overall race time performances in the Ironman World Championship from 1983 to 2018 of women and men. Gold trophies represent the champion in each year.

**Figure 2 ijerph-16-01019-f002:**
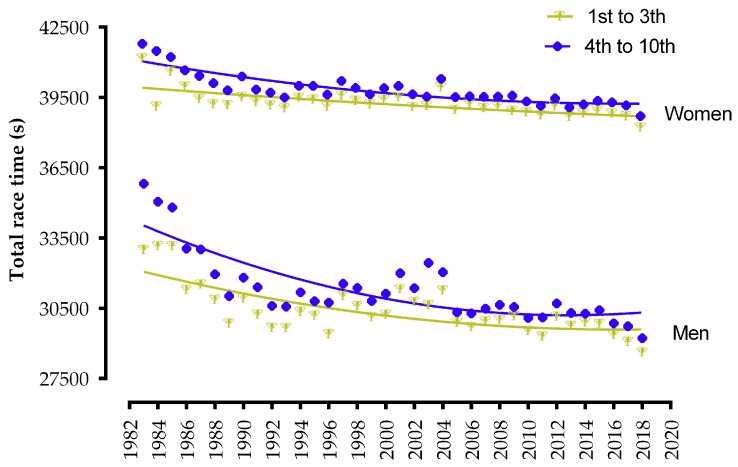
Dispersion and non-linear regression overall race time performances between the top three finishers and the chasing group (4th to 10th place finishers) from women and men in the Ironman World Championship from 1983 to 2018.

**Figure 3 ijerph-16-01019-f003:**
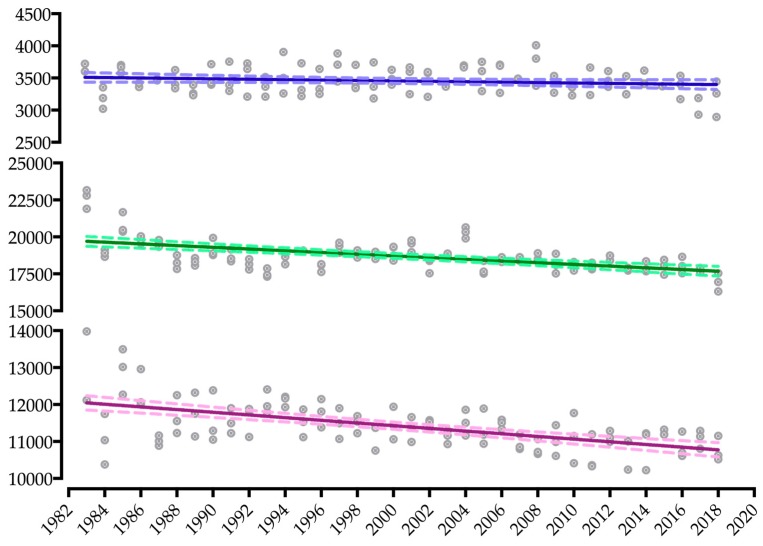
Dispersion and linear regression of split-times performances in the Ironman World Championship from 1983 to 2018 of women.

**Figure 4 ijerph-16-01019-f004:**
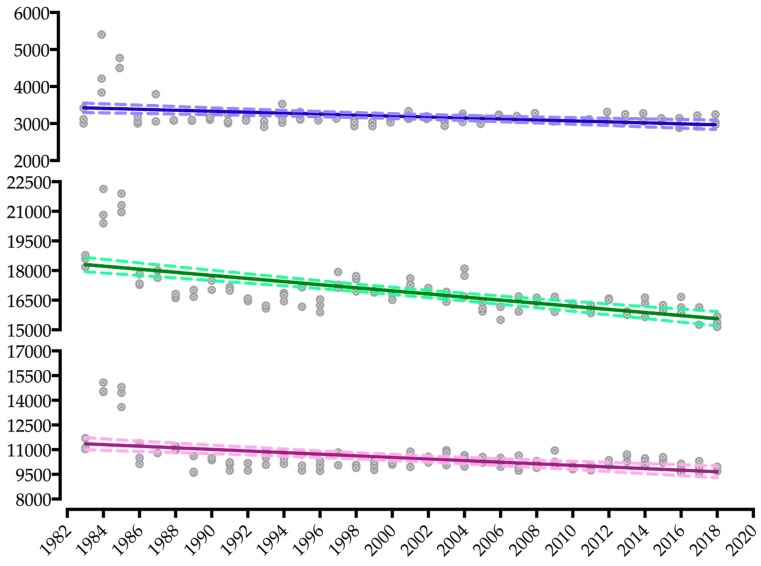
Dispersion and linear regression of split-times performances in the Ironman World Championship from 1983 to 2018 of men.

**Table 1 ijerph-16-01019-t001:** Women’s total and split race times in the Ironman World Championship from 1983 to 2018.

Race Time	Median (25–75 Percentile)	Mean (±SD)	Ironman World Championship Race/Split Record *
Overall	09:16:48 (09:03:51–09:26:18)	09:19:06 (00:26:14)	08:26:18
Swimming	00:57:00 (00:55:26–01:00:09)	00:57:34 (00:03:24)	00:48:14
Cycling	05:08:39 (05:00:14–05:17:50)	05:11:22 (00:18:06)	04:26:07
Running	03:08:10 (03:04:09–03:16:31)	03:10:10 (00:10:37)	02:50:26

* Within top three finishers from 1983 to 2018.

**Table 2 ijerph-16-01019-t002:** Men’s total and split race times in the Ironman World Championship from 1983 to 2018.

Race Time	Median (25–75 Percentile)	Mean (±SD)	Ironman World Championship Race/Split Record *
Overall	08:22:02 (08:14:37–08:33:02)	08:26:28 (00:18:26)	07:52:39
Swimming	00:51:43 (00:51:00–00:53:02)	00:53:18 (00:06:07)	00:48:02
Cycling	04:37:47 (04:30:16–04:46:15)	04:42:15 (00:21:01)	04:12:25
Running	03:08:10 (02:46:42–02:57:00)	02:55:21 (± 00:18:57)	02:39:59

* Within top three finishers from 1983 to 2018.

**Table 3 ijerph-16-01019-t003:** Women’s and men’s percentage performance improvements in the Ironman World Championship from 1983 to 2018.

		Total	Total Difference	Decade Average	Decade Average Difference
Overall	Women	20.8%	7.5%	5.20%	1.87%
	Men	13.3%		3.33%	
Swimming	Women	12.1%	9.1%	3.25%	2.50%
	Men	3%		0.75%	
Cycling	Women	26.4%	9.5%	6.60%	2.37%
	Men	16.9%		4.23%	
Running	Women	15.5%	3%	3.88%	0.75%
	Men	12.5%		3.13%	

**Table 4 ijerph-16-01019-t004:** Standardized coefficient from stepwise multiple regression using total race time as the dependent variable of Ironman World Championship from 1983 to 2018.

	Standardized *β* Coefficient			*R* ^2^	*R* ^2^ _aj_
	Swimming	Cycling	Running		
Women	0.129	0.690	0.405	0.857	0.856
Men	−0.290	0.895	0.250	0.781	0.775

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
