# Peer review of "Celebrating 40 Years of Ironman: How the Champions Perform"

_ijerph, 2019, doi:10.3390/ijerph16061019_

Round 1

Reviewer 1 Report

Barbosa and colleagues present an analysis of the Ironman Hawaii top 3 men and women finishing times of the last 35 years. They are aiming to answer three main questions: i) performance development overall and within the three disciplines (swimming, cycling, running); ii) magnitude of influence of the single disciplines on overall race results; iii) differences in gender.

The analysis yields some small insights into the development of the overall performance within the Ironman World Championships. Even though the aims and the purpose of the study are interesting, from my perspective the data set used for this analysis is to weak to draw profound conclusions. To only analyze the top 3 men and women does not provide a solid basis for the raised questions. Why weren´t the top 10 men and women included into the analysis (as some kind of "control group" to the winners)?

The manuscript needs a major editing for language and consistency. Many typos and carelessnesses are evident.

The discussion is lacking of comparisons to performance development to other triathlon distances and/or in other endurance disciplines. This makes the discussion very weak.

Minor:

Line 51: Please provide more information about the other triathlon races you mention here (duration and/or distances). E.g. I guess half distance means half Ironman, full distance means Ironman, and for me ultra-triathlons are longer than Ironman and therefore the second part of the sentence makes no sense to me: “… comprehending distances and races longer than the traditional Ironman distance…”.

Table 1 & 2: What is meant by World Record here? This is the Hawaii Championship record, and not World best time, isn´t it?

Figure 1: Even though I like the trophies, this symbol makes it difficult to identify the exact value of the champion´s time.

Figure 2 & 3: The slope of the linear regressions would be helpful.

Author Response

REVIEWER 1

INITIAL COMMENT: Barbosa and colleagues present an analysis of the Ironman Hawaii top 3 men and women finishing times of the last 35 years. They are aiming to answer three main questions: i) performance development overall and within the three disciplines (swimming, cycling, running); ii) magnitude of influence of the single disciplines on overall race results; iii) differences in gender.

COMMENT #1: The analysis yields some small insights into the development of the overall performance within the Ironman World Championships. Even though the aims and the purpose of the study are interesting, from my perspective the data set used for this analysis is to weak to draw profound conclusions. To only analyze the top 3 men and women does not provide a solid basis for the raised questions. Why weren´t the top 10 men and women included into the analysis (as some kind of "control group" to the winners)?

RESPONSE #1: We agree with the expert reviewer and we are grateful for the comment. A new comparison chart (Figure 4) were made between the first group (top 3) and a second group (4th to 10th). Also see the added text in methods and results sections highlighted in red. However, we invite the esteemed reviewer to consider that the inclusion of only the first three athletes in each race is in accordance with the objectives of the present investigation. Thus, it is reasonable to assume that the fact that we initially selected only those does not seem to weak the findings.

COMMENT #2: The manuscript needs a major editing for language and consistency. Many typos and carelessnesses are evident.

RESPONSE #2: The manuscript was sent again to a native English speaking for a new language review and typos removal. Thank you for the comment.

COMMENT #3: The discussion is lacking of comparisons to performance development to other triathlon distances and/or in other endurance disciplines. This makes the discussion very weak.

RESPONSE #3: Dear reviewer, we discussed our results with Olympic, short and ultra-triathlon distances using the best available literature, please see paragraph 4 discussion. However, it is worth emphasizing that the discussion section should explain the meaning of the results to the reader and not simply compare the findings with other studies (Hess, 2004). Thus, we invite the esteemed reviewer to consider that this aspect does not configure in a fragility of the present study, but rather that we only tried to accomplish what is expected in a discussion section.

Hess, D. R. (2004). How to write an effective discussion. Respiratory care, 49(10), 1238-1241.

COMMENT #4: Line 51: Please provide more information about the other triathlon races you mention here (duration and/or distances). E.g. I guess half distance means half Ironman, full distance means Ironman, and for me ultra-triathlons are longer than Ironman and therefore the second part of the sentence makes no sense to me: “… comprehending distances and races longer than the traditional Ironman distance…”

RESPONSE #4: We agree with the reviewer, and we added the information required. Thank you for the comment. Please see changes highlighted in red.

“...Specifically in triathlon, relevant studies have been conducted at Olympic (1.5 km swim/ 40 km cycle/ 10 km run) [12,13], half distance (half - Ironman: 1.9 km swim/ 90 km cycle/ 21 km run) [13,14], full distance [15,14], and ultra-triathlons (distance > Ironman) [16,17].”

COMMENT #5: Table 1 & 2: What is meant by World Record here? This is the Hawaii Championship record, and not World best time, isn´t it?

RESPONSE #5: Thanks for the comment. We changed the name to “Ironman World Championship Race/Split Record”

COMMENT #6: Even though I like the trophies, this symbol makes it difficult to identify the exact value of the champion´s time.

RESPONSE #6: We added a new Supplementary Table with the exact champion’s race time in each year.

COMMENT #7: The slope of the linear regressions would be helpful.

RESPONSE #7: We agree. The slope of the linear regressions has been placed as suggested.

Reviewer 2 Report

I really enjoyed the manuscript: it is very original and it is a solid analysis of a booming sport like triathlon is. Only two formal details to take into account:

- Page 3, line 90, I guess is "than", instead of "then".

- Please check references, as sometimes the name of the journal is abbreviatted, other not; sometimes all letters in capital, others just the first. Please unify.

Author Response

REVIEWER 2

INITIAL COMMENT: I really enjoyed the manuscript: it is very original and it is a solid analysis of a booming sport like triathlon is. Only two formal details to take into account:

COMMENT #1: Page 3, line 90, I guess is "than", instead of "then".

RESPONSE #1: We appreciate the suggestion and made the changes as suggested.

COMMENT #2: Please check references, sometimes the name of the journal is abbreviatted, other not; sometimes all the letters in capital, others just the first. Please unify.

RESPONSE #2: We carefully check the manuscript references to correct all inaccuracies.

Round 2

Reviewer 1 Report

Thank you for the revised version of the manuscript. I appreciate to see that the results of the 4th – 10th place have been included. Please add a small part in the discussion as well. From a “visual analysis” of figure 2 it seems that the density in the top 10 became higher in the 30 years of IM Hawaii?!

Minor:

Line 47: Kailua-Kona

Line 52: )

Line 53f: Please specify. This sentence lacks of information with regards to the sentences before. The references are explained later in lines 62ff. Perhaps the sentence in lines 53f can be deleted.

Line 78: “World Record” please use “Ironman World Championship Race/Split Record” to be consisted with the tables.

Table 4: I´m not sure if I understand the table correctly. You say only swimming has a negative standardized coefficient for the men. Why is the coefficient for the men´s cycling negative as well?

Line 133: finding instead of findings

Line 134: influence on

Line 137-138: Jeukendrup & Martin (without first names)

Author Response

REVIEWER 1

INITIAL COMMENT: Thank you for the revised version of the manuscript. I appreciate to see that the results of the 4th – 10th place have been included. Please add a small part in the discussion as well. From a “visual analysis” of figure 2 it seems that the density in the top 10 became higher in the 30 years of IM Hawaii?!

RESPONSE: We thank the expert reviewer for all the comments. A small part of regarding the 4th – 10th place was added in the discussion, as suggested. Please see changes highlighted in red.

COMMENT #1: Line 47: Kailua-Kona

RESPONSE #1: We appreciate the suggestion and made the changes as suggested. Please see changes highlighted in red.

COMMENT #2: Line 52: )

RESPONSE #2: Thanks for observation, we corrected it

COMMENT #3: Line 53f: Please specify. This sentence lacks of information with regards to the sentences before. The references are explained later in lines 62ff. Perhaps the sentence in lines 53f can be deleted.

RESPONSE #3: We reconsidered the sentence in that part of the manuscript and agreed with the expert reviewer. We therefore moved it to end of the paragraph.

COMMENT #4: “World Record” please use “Ironman World Championship Race/Split Record” to be consisted with the tables.

RESPONSE #4: We agree with the reviewer, and we change the information required. Thank you for the comment. Please see changes highlighted in red.

COMMENT #5: I´m not sure if I understand the table correctly. You say only swimming has a negative standardized coefficient for the men. Why is the coefficient for the men´s cycling negative as well?

RESPONSE #5: Dear reviewer, the negative symbol in the Table was a typo. We apologize for the confusion and thank you for this very important comment.

COMMENT #6: finding instead of findings

RESPONSE #6: thanks for the comment, change as suggested

COMMENT #7: influence on

RESPONSE #7: thanks for the comment, change as suggested

COMMENT #8 Jeukendrup & Martin (without first names)

RESPONSE #8: thanks for the comment, change as suggested